# OpenReview forum: "Whose View of Safety? A Deep DIVE Dataset for Pluralistic Alignment of Text-to-Image Models"
_NeurIPS.cc/2025/Datasets_and_Benchmarks_Track — NeurIPS 2025 Datasets and Benchmarks Track spotlight_

### Official Review · Reviewer_323P · 2025-06-23

**Ethics Flags:** Safety and security
**Rating:** 5
**Confidence:** 3

**Summary:**

This paper addresses a critical issue in text-to-image (T2I) model evaluation, i.e., the subjective and culturally contingent nature of safety. It proposes pluralistic alignment as a solution, emphasizing that AI safety should incorporate diverse human perspectives. To achieve this, the authors introduce the Diverse Intersectional Visual Evaluation (DIVE) dataset, specifically crafted to facilitate pluralistic alignment. DIVE is a multimodal dataset featuring extensive annotations from 637 demographically diverse raters across 30 intersectional demographic categories (gender, age, ethnicity). The dataset consists of 1,000 prompt-image pairs, classified across three violation categories (Explicit, Violent, and Bias) and 12 thematic topics, generating over 35,000 harm evaluations. Additionally, the paper empirically demonstrates the significance of demographic intersectionality in shaping harm perception, and discusses implications for developing more equitably aligned T2I models, including the feasibility of employing Large Language Models (LLMs) as evaluators and steerable models.

**Dataset Code Accessibility:**

Yes

**Ethical Comments:**

The potential psychological impact on subjects resulting from repeated exposure to harmful content should be taken into consideration.

**Ethical Considerations:**

Yes, there are ethics concerns that require attention by the authors

**Final Justification:**

Thank you for the author's rebuttal. The provided discussions, analyses, and experiments have partially addressed my concerns. I believe this work presents a topic worthy of exploration. Future research on more diversified scenarios is highly anticipated, and this includes the alignment of generative models.

Meanwhile, I respect and understand that alignment of generative models is outside of the scope of D&B track, so I have considered raising my score from 4 to 5. Meanwhile, I still hope to see relevant explorations and attempts.

**Limitations Weaknesses:**

1.	While the paper acknowledges the relatively weak performance of the baseline LLMs used for evaluations, there is room for deeper analysis or experimentation with more powerful state-of-the-art multimodal models. The presented LLM analysis, although insightful, leaves open questions about the effectiveness and scalability of this approach.
2.	Despite efforts to mitigate distress and bias during annotation, the psychological impacts of repeated exposure to harmful content remain a concern. More detailed analysis of annotation fatigue or bias, including checks for temporal effects across sessions, would strengthen validity.
3.	Although carefully curated, the 1,000 prompt-image pairs may still limit the generalizability and comprehensiveness of harm evaluations. Expanding or diversifying the sample set further could better capture rare but critical edge-cases of harm.
4.	While the paper effectively establishes demographic intersections (gender, age, ethnicity) as predictive of harm perception, it somewhat neglects other potential demographic dimensions such as socio-economic status, education level, or cultural background beyond ethnicity. Expanding this scope could further enrich the dataset's pluralistic alignment capabilities.
5.	Although the authors claim that this work can optimize subsequent text-to-image models, could the authors provide some concrete results to support this assertion? For example, have they attempted to fine-tune Stable Diffusion (SD) based on this work?

**Strengths Contributions:**

1.	The paper clearly addresses an important and currently underexplored dimension in AI alignment, the plurality of human safety perspectives, particularly critical for T2I generative models, where interpretations of harm are deeply subjective and culturally specific.
2.	The proposed DIVE dataset is groundbreaking as the first multimodal resource dedicated explicitly to pluralistic alignment.
3.	Empirical analyses are robust, clearly demonstrating that intersectional demographic groupings significantly influence harm perception.
4.	The findings have direct and practical implications for future T2I model

---

> ### Author Rebuttal · Authors · 2025-07-30
>
> We are pleased to see that the reviewer appreciated the importance of the problems we are addressing, the novelty of the DIVE dataset and the robustness of the empirical methodology. We also appreciate you taking the time to provide extra feedback to ensure our results and overarching message are communicated with maximum clarity and impact. Below we provide detailed responses to your questions and concerns:
>
> ## 1: Analysis of LLMs for safety evaluation and experimenting with larger models
> Thank you for raising this. To have a more comprehensive evaluation, we conducted **new experiments with a larger model family** (Gemini 1.5 Flash distilled from a 200B parameter Gemini 1.5 Pro model) for each strategy - (1) prompting with demographic information
> and (2) prompting with demographic information plus few shot examples from DIVE.
>
> ****Our new experiments with a larger model show correlation improvement with few shot examples, showcasing steerability****
>
> Now, we provide the Kendall-Tau correlation of the diverse raters and the LLM-based rater averaged for each top level demographic (in parenthesis):
>
> **(1) Gemini1.5F prompted with demographic information (extension of Table 2 in paper)** KT correlation:
> - 0.25 (Men), 0.26 (Women),
> - 0.25 (GenX), 0.25 (Millennial), 0.26 (GenZ),
> - 0.23 (Black), 0.24 (Eastasian) 0.26 (Latinx), 0.27 (Southasian), 0.27 (White)
>
> **(2) Gemini1.5F prompted with demographic information and few-shot examples** KT correlation:
> - 0.28 (Men), 0.29 (Women),
> - 0.28 (GenX), 0.28 (Millennial), 0.30 (GenZ),
> - 0.27 (Black), 0.27 (Eastasian) 0.28 (Latinx), 0.30 (Southasian), 0.30 (White)
>
> **Insights:** We see that the larger model when provided demographic information and few-shot examples performs better than the other model (Gemma) and other strategies considered. Interestingly **providing examples from DIVE improves the metric with the Gemini model, showcasing steerability.** However, the correlations remain modest, thus emphasising the importance of DIVE for future research on methods to develop LLM-based raters in the T2I space. We will discuss this result in further detail in the paper breaking down the results by trisections, to convey the extent of the gap in representation of different identities with LLM-based approaches to evaluation.
>
> **Updates to paper narrative**: The goal of this experiment is to understand the (lack of) representativeness of different demographic identities by LLM-based autoraters using baseline methods for steering. We will update the writing of Section 5.2 to reflect that and further emphasise the takeaways for pluralistic alignment research.
> Relatedly, we note here that LLM-as-judge style model alignment via algorithm design and experimentation is out of scope for this work, in keeping with the D&B track goals, and outlined as future work in our manuscript.
>
> ## 2: Analysis of annotation fatigue or bias, checks for temporal effects across sessions
> Thank you for raising this point. To ensure high quality data and analyses, we filtered out participants whose responses contained inconsistencies, low accuracy on attention checks, and other undesirable behaviour. We laid out our detailed approach for this in App A.4. Common indicators of annotation fatigue were taken into account when designing our quality filtering approach. While we acknowledge that temporal effects can be a factor, given that each participant in our study engaged with the task only around 30-45 mins, the temporal effects such as fatigue are likely limited.
>
>
> ## 3: Larger dataset to capture rare but critical edge-cases of harm.
>
> **Diversity within Prompt-Image-pair set**:  As described in Section 3, when choosing the 1000 PI-pair set, several important considerations were taken into account to ensure that the PI pairs are diverse, showcase a variety of harms, and are suitable for understanding pluralistic alignment. This ensures that the dataset has useful and actionable implications for safety evaluation. We list the considerations here:
> - The dataset is diverse containing three different harm types each with sizeable representation
> - The dataset contains PI pairs concerning unique and varied topics each with uniform representation
>
>
> **Prompt-image pair source (Adversarial Nibbler dataset)**: Lastly, the PI pairs were sourced from Adversarial Nibbler dataset [3] which had two unique and important features making it a high quality dataset containing many edge cases of harm (as detailed in Section 3):
> - AdvNib Challenge participants were instructed to provide implicitly adversarial prompts, which resulted in a prompt set well positioned to uncover novel safety blindspots [3].
> - Each PI pair in AdvNib was annotated by five policy raters and we chose those with the highest disagreement to prioritise ambiguous cases of harm.
>
> **Depth vs breadth trade-off**
> - The DIVE dataset has a unique focus on understanding the safety perspectives of 30 unique demographic trisections on each PI pair. Thus, we take a depth-first approach to safety evaluation by trading off a higher PI-pair count in favor of ratings from more unique demographic identities.
> - The importance of this is evidenced by the findings in Section 4 and 5, where we see that some demographically intersectional groups are more cohesive, different intersectional groups identify harms differently in different prompt-Image types, and current safety evaluation methods show variance in their representativeness of different demographics.
> - The high number of ratings enable statistically rigorous analysis, as previously discussed in [1].
> - In addition, past research dataset [2] for pluralistic alignment with a focus on diverse annotation has a similar dataset size.
>
> ## 4: Expanding with socio-economic status, education, or culture to enrich the dataset's pluralistic alignment capabilities.
> We agree that these other demographic dimensions are important axes to consider for pluralistic alignment. In our analyses, we primarily focus on gender, age, and ethnicity for two reasons.
> - First, aligning with prior works on social biases  and safety perception [2] which frequently focus on gender, race, and age, we collected a relatively balanced annotator sample across these dimensions.
> - To ensure statistical power in our analysis, we did not stratify our recruitment across all possible demographic variables such as class and education status.
> - Lastly, cultural background was difficult to assess beyond a focus on race and ethnicity due to recruitment in the US and the UK. Cultural background, in particular, is a rich area for exploration in future iterations of this work.
>
> ## 5: Showing this work can optimize subsequent text-to-image models
>
> Thank you for raising this important point. We want to clarify that the scope of the paper with the following updates to writing:
>
> **Updates to writing**
>
> - **Framing**: We will clearly frame the narrative to clarify that the point you raised **is not an oversight but an important insight aligned with the goals of the paper**:
>     - The **core contribution of this paper is the curated, pluralistic dataset**, intended to provide a valuable benchmark for the research community.
>     - Our **experiments serve as a baseline analysis** to demonstrate the dataset's utility and provide a necessary "headroom analysis" of current model capabilities
>
> - **Future work section**: Lastly, following your feedback, in the future work section we will outline clearly the **specific next steps towards operationalizing this dataset for improving pluralistic alignment in T2I models**:
>     - DIVE can be used for model training and fine-tuning and evaluating LLMs’ ability to replicate diverse safety perspectives, as has been done in past work building upon T2T safety datasets [4,5].
>     -  DIVE provides rich preference data via granular scoring which is useful for RLHF-based experiments on steering models towards alignment with diverse raters [6], which is a large and open research problem. Many approaches in reward modeling [7] and mixed experts [8] would benefit from the diverse viewpoints in DIVE.
>
>
> **Alignment with D&B track goals:** We would like to note (lines 319-333) that our DIVE dataset contributes in the same space as prior research in text-to-text datasets, published in the NeurIPS Dataset & Benchmarks track. These works similarly identified gaps in safety evaluations and published datasets that have been since used to improve model alignment (without providing a model-based solution to alignment) [4,5]. Examples of such papers are : “The PRISM Alignment Dataset” by Kirk et al, 2024 (best D&B paper in 2024) and DICES dataset [2].
>
> Both these papers focus on T2T issues, and our dataset builds upon them with more advanced statistical design and uniquely rich data into the T2I space. Furthermore, the D&B track goals are to stimulate research papers introducing datasets that help evaluate and benchmark models. Similarly, the scope of DIVE is to show the variety of gaps and issues caused by not considering diverse viewpoints crucial to be addressed as part of pluralistic alignment.
>
> **References**
>
> [1] How Many Raters Do You Need? Power Analysis for Foundation Models. Homan et al 2023
>
> [2] DICES Dataset: Diversity in Conversational AI Evaluation for Safety. Aroyo et al 2023.
>
> [3] Adversarial Nibbler: An Open Red-Teaming Method for Identifying Diverse Harms in Text-to-Image Generation. Quaye et al 2024
>
> [4] How Far Can We Extract Diverse Perspectives from Large Language Models? Hayati et al 2024
>
> [5] LLMs instead of Human Judges? A Large Scale Empirical Study across 20 NLP Evaluation Tasks. Bavaresco et al 2025
>
> [6] Personalizing Reinforcement Learning from Human Feedback with Variational Preference Learning. Poddar et al 2024
>
> [7] MaxMin-RLHF: Alignment with Diverse Human Preferences. Chakraborty et al 2024
>
> [8] Collab: Controlled decoding using mixture of agents for llm alignment Chakraborty et al 2025

---

> > ### Comment · Reviewer_323P · 2025-08-01
> > **Official Comment by Reviewer 323P**
> >
> > Thank you for the author's rebuttal. The provided discussions, analyses, and experiments have partially addressed my concerns. I believe this work presents a topic worthy of exploration. Future research on more diversified scenarios is highly anticipated, and this includes the alignment of generative models. Therefore, I decide to maintain my score.

---

> > > ### Author Response · Authors · 2025-08-01
> > > **Seeking more engagement, clarification, and reasoning for decision despite highly positive assessments in review**
> > >
> > > Dear Reviewer,
> > >
> > > Thank you for your response to our rebuttal. We appreciate you taking the time to engage with our work.
> > >
> > > We are particularly encouraged by **your highly positive assessment in your review**, where you highlighted the groundbreaking nature of our dataset, the robustness of our studies, and the practical implications of our findings. You noted:
> > > - _"The proposed DIVE dataset is groundbreaking as the first multimodal resource dedicated explicitly to pluralistic alignment."_
> > > - _"Empirical analyses are robust, clearly demonstrating that intersectional demographic groupings significantly influence harm perception."_
> > > - _"The findings have direct and practical implications for future T2I model."_
> > >
> > > Given this strong praise, we believe the contributions of our paper extend beyond a "borderline accept" as overall score. We would welcome meaningful discussion about your decision to maintain the overall score, as we are confident more discussion can resolve any remaining reservations.
> > >
> > > Your recent feedback states that we have only **"partially addressed" your concerns,
> > > Could you please specify which aspects of your questions were resolved and which require further clarification.** This would also facilitate the completion of the second part of the rebuttal acknowledgement: "I have engaged in discussions and responded to authors."
> > >
> > > To provide a summary of our position discussed in detail in our rebuttal:
> > >
> > > - **Regarding Questions #1 and #5:** We maintain that these points are outside the scope of this paper and the expectations from the CFP for dataset papers (as detailed in our rebuttal).
> > >
> > > Despite this, **we conducted new experiments with the Gemini 1.5 model family**, exploring two different prompting strategies and demonstrating / confirming steerability with few-shot examples from DIVE. We are really keen to understand which parts of these questions you feel remain unaddressed and why.
> > >
> > > - **Regarding Questions #2, #3, and #4:** We would also appreciate the opportunity to discuss these points further to understand which aspects you found satisfactory and where you believe our responses fell short.
> > >
> > > We look forward to a constructive and clarifying discussion so we can fully address your concerns and further strengthen our manuscript. Thank you in advance for your continued engagement

---

> > > > ### Comment · Reviewer_323P · 2025-08-02
> > > > **Official Comment by Reviewer 323P**
> > > >
> > > > I hold a positive view of this work and believe it meets the acceptance criteria. However, I still hope to see some attempts regarding the alignment of generative models. Similar endeavors are also included in works like ImageReward, which I consider meaningful.

---

> > > > > ### Author Response · Authors · 2025-08-04
> > > > > **Response to reviewer**
> > > > >
> > > > > Dear Reviewer,
> > > > >
> > > > > Thank you again for your response and saying **our work meets the acceptance criteria**. Given this response, we are hopeful you will update the score accordingly.
> > > > >
> > > > > To facilitate a more productive discussion, could you please elaborate on which of our specific points raised in your review that are **not related to alignment work** (Questions #2, #3, and #4) are sufficiently addressed by our rebuttal, and which are the remaining points that remain unaddressed?
> > > > >
> > > > > _Regarding alignment work:_ We would like to **reiterate once again that the request for more alignment research is outside of the scope of our paper and the scope of the D&B track** (as we motivated and clarified both in the rebuttal and in the follow up response).  This paper introduced a dataset and the unique properties it offers (including data collection method). We believe that holding us responsible for aspects outside of the scope of the D&B track (and the defined scope of the paper) may not be conducive to a fair evaluation of our contribution (especially when compared with other papers in the D&B track). In this context, we believe that the reference to ImageReward in your final comment is misleading - as it is a Reward Modelling approach paper **(accepted in the Main track, not the D&B track)**.
> > > > >
> > > > > Thank you for your time and consideration.

---

> > > > > > ### Comment · Reviewer_323P · 2025-08-05
> > > > > > **Official Comment by Reviewer 323P**
> > > > > >
> > > > > > Thank you for your response. I respect and understand that alignment of generative models is outside of the scope of D&B track, so I have considered raising my score from 4 to 5. Meanwhile, I still hope to see relevant explorations and attempts.

---

### Official Review · Reviewer_cdP4 · 2025-06-28

**Rating:** 5
**Confidence:** 2

**Summary:**

This paper introduces diverse intersectional visual evaluation (DIVE), a multimodal dataset for pluralistic alignment of text-to-image (T2I) models. The dataset contains 31980 human evaluations of 1000 prompt-image pairs from 637 raters across 30 demographic intersections. The authors demonstrate that demographic attributes significantly influence harm perception in T2I content, with intersectional groups showing higher cohesion than broader demographic categories. They explore implications for building aligned T2I models through diverse feedback collection and LLM steering experiments.

**Additional Feedback:**

No more comments.

**Dataset Code Accessibility:**

Yes

**Dataset Code Comments:**

The dataset is publicly available on HuggingFace with detailed documentation. The experimental procedures are well-documented with sufficient detail for reproduction.

**Ethical Considerations:**

No, there are no or only very minor ethics concerns

**Final Justification:**

Thanks for the author's rebuttal. Among these three responses, I believe the second point regarding prompt and image safety effects is the most convincing. The author has addressed my concerns, and I will keep my score.

**Limitations Weaknesses:**

1. Table 2 shows modest correlations between steered LLMs and human ratings, barely above baseline. The authors acknowledge this, but the implications for practical pluralistic alignment remain unclear. The choice of 4B parameter models seems inadequate for this complex task.

2. As noted in the limitations, the study cannot disentangle prompt safety from image safety effects. This may be a significant methodological limitation that affects the interpretation of results, particularly given the adversarial nature of the source dataset.

3. The attention check procedures for identifying low-quality raters could be better detailed.

**Strengths Contributions:**

1. The intersectional demographic sampling strategy is more sophisticated than typical high-level demographic groupings. This design enables granular analysis of how overlapping identities shape harm perception, which is methodologically sound and practically important.

2. The paper provides compelling evidence that demographics matter for safety evaluation.

3. The simulation studies (Figure 3) showing differential flagging patterns across demographic groups provide actionable insights about whose perspectives might be missed in typical evaluations.

---

> ### Author Rebuttal · Authors · 2025-07-30
>
> ## Reviewer comment 1: Table 2 shows modest correlations between steered LLMs and human ratings, barely above baseline. The authors acknowledge this, but the implications for practical pluralistic alignment remain unclear. The choice of 4B parameter models seems inadequate for this complex task.
>
> Thank you for raising this. **We conducted experiments with a larger model family (Gemini 1.5 Flash distilled from a 200B parameter Gemini 1.5 Pro model)** for each strategy - (1) prompting with demographic information and (2) prompting with demographic information plus few shot examples from DIVE.
>
> ****Our new experiments with a larger model show correlation improvement with few shot examples, showcasing steerability****
>
> Specifically, here we provide the Kendall-Tau correlation of the diverse raters and the LLM-based rater averaged for each top level demographic (in parenthesis):
>
> **(1) Gemini1.5F prompted with demographic information (extension of Table 2 in paper)**
> KT correlation:
> - 0.25 (Men), 0.26 (Women),
> - 0.25 (GenX), 0.25 (Millennial), 0.26 (GenZ),
> - 0.23 (Black), 0.24 (Eastasian) 0.26 (Latinx), 0.27 (Southasian), 0.27 (White)
>
> **(2) Gemini1.5F prompted with demographic information and few-shot examples**
> KT correlation:
> - 0.28 (Men), 0.29 (Women),
> - 0.28 (GenX), 0.28 (Millennial), 0.30 (GenZ),
> - 0.27 (Black), 0.27 (Eastasian) 0.28 (Latinx), 0.30 (Southasian), 0.30 (White)
>
> **Insights:** We see that the larger model when provided demographic information and few-shot examples performs better than the other model (Gemma) and other strategies considered. Interestingly **providing examples from DIVE improves the metric with the Gemini model, showcasing steerability.** However, the correlations remain modest, thus emphasising the importance of DIVE for future research on methods to develop LLM-based raters in the T2I space. We will discuss this result in further detail in the paper breaking down the results by trisections, to convey the extent of the gap in representation of different identities with LLM-based approaches to evaluation.
>
> **Scope of the paper:** Thank you for raising this point, as it also reflects on the scope of the paper.
> - We would like to note here that **LLM-as-judge style model alignment via model development and experimentation is out of scope for this work**, in keeping with the D&B track goals, and outlined as future work in our manuscript.
> - The **experiments presented in the paper serve as a key and necessary analysis** to demonstrate the dataset's utility and provide  a "headroom analysis" of current model capabilities in pluralistic alignment.
> - We will revise Section 5.2 to clarify this goal and de-emphasise the focus on “steering LLMs for pluralistic alignment” and focus the narrative on steerability of LLMs using off-the-shelf methods for auto-evaluation of safety.
>
>
> ## Reviewer comment 2: As noted in the limitations, the study cannot disentangle prompt safety from image safety effects. This may be a significant methodological limitation that affects the interpretation of results, particularly given the adversarial nature of the source dataset.
>
> While we acknowledge this limitation, we did take steps to mitigate its effect on the data. Notably, our instructions clearly indicated that the raters should base their judgments only on the image content, rather than the content of the prompt. Prompts were shown to raters to help disambiguate or provide context in examples such as those concerning stereotyping behaviour. Additionally, the rating task specifically asked raters how safe the **image** was to them or others to reinforce the instructions to base the safety rating on characteristics of the image rather than the prompt.
>
> Next, it is important to note that the source prompt set Adversarial Nibbler [1] was designed to have “implicitly adversarial prompts”, that is the participants of the Adversarial Nibbler challenge were specifically instructed to provide prompts that are textually safe and yet yield harmful image content.
>
> Thus, with these two points in mind, our work takes considerable steps in mitigating the impact of prompt safety on the data. We will add these points to the updated manuscript to clarify the point raised.
>
>
> ## Reviewer comment 3: Better detailing the attention check procedures for identifying low-quality raters
> Thank you for raising this point; we have detailed all the steps we followed for identifying low-quality raters in Appendix A.4 of the submitted manuscript. We provide here an abridged version for your perusal. Given this context, it would be very helpful to know whether there are any specific aspects that the reviewer considers missing and crucial to address.
>
> **Low-Quality filtering procedure:** To ensure high-quality data, we implemented a multi-stage rater quality control process. Initially, we filtered out raters who completed fewer than 45 total annotations or failed more than one of the five embedded attention-check questions.
> Surviving raters were then automatically flagged for manual review if they met certain criteria, such as low attention check accuracy, low time spent on the task, infrequent comments, high response inconsistency, or an unusually high number of "Not harmful" ratings.
> A detailed manual inspection followed for all flagged raters. During this review, we examined their specific responses for signs of low effort, such as unjustified errors on attention checks, patterned or formulaic answers, nonsensical violation classifications, and consistently fast response times. Based on this comprehensive review, a final decision was made to either keep or discard the rater's data.
>
> **References**
> [1] Adversarial Nibbler: An Open Red-Teaming Method for Identifying Diverse Harms in Text-to-Image Generation. Quaye et al 2024

---

> > ### Comment · Reviewer_cdP4 · 2025-08-01
> >
> > Thanks for the author's rebuttal. Among these three responses, I believe the second point regarding prompt and image safety effects is the most convincing. The author has addressed my concerns, and I will keep my score.

---

> > > ### Author Response · Authors · 2025-08-01
> > > **Seeking meaningful engagement on rebuttal and clarification on reasoning**
> > >
> > > Dear Reviewer,
> > > Thank you for your acknowledgment of our rebuttal and for finding our response to Question #2 satisfactory. We are grateful for your time and the feedback you have provided throughout this process.
> > >
> > > We are particularly encouraged by **your positive assessment of our work's strengths** in your initial review.
> > > - You noted that our _"intersectional demographic sampling strategy is more sophisticated than typical high-level demographic groupings"_ and that our design _"enables granular analysis of how overlapping identities shape harm perception, which is methodologically sound and practically important."_
> > > - You commended the paper for providing _"compelling evidence that demographics matter for safety evaluation"_ and that the "_simulation studies (Figure 3) showing differential flagging patterns across demographic groups provide actionable insights about whose perspectives might be missed in typical evaluations._"
> > >
> > > Thus, given your extremely positive evaluation of our paper's methodology, novelty, and results, combined with the rebuttal providing new experiments and fully addressing all your concerns, we were optimistic this would be sufficient to raise the score. We are keen to engage in a constructive discussion about your decision to maintain it at 'Borderline accept.'"
> > >
> > > Next, could you please **provide specific feedback** regarding your assessment of our responses to Questions #1 and #3, as your current response does not specify
> > > (1) which aspects of the responses you did not find convincing, or
> > > (2) what would be needed to fully address your concerns.
> > >
> > > **Regarding Question #1**: we would like to respectfully reiterate our efforts to address your question. Although it is **outside the scope of our paper as well as the call for dataset papers**, to address your concerns, we **conducted new experiments** with the Gemini 1.5 model family. We also shared results on the improved correlation with few-shot examples, demonstrating / confirming steerability with larger models. We are keen to understand which specific aspects of your question remain unaddressed.
> > >
> > > **Regarding Question #3:** we would again like to highlight that the requested information was present in the original manuscript (App A.4). In our rebuttal, we provided direct references to its location and a summary of the attention check procedure to facilitate your review. We welcome clarification on what specific points in this question you find is not sufficiently addressed in our rebuttal.
> > >
> > > Finally, **we are eager to engage in a constructive dialogue** to resolve these outstanding points. We believe a more detailed and engaged discussion will **allow for the completion of the second part of the rebuttal acknowledgement: "I have engaged in discussions and responded to authors."**

---

> > > > ### Author Response · Authors · 2025-08-04
> > > > **Another attempt in seeking meaningful engagement on rebuttal and clarification**
> > > >
> > > > Dear reviewer,
> > > >
> > > > Thank you for acknowledging our rebuttal.
> > > >
> > > > We would be grateful if you could engage with us in a constructive dialog and provide more detailed comments on how our rebuttal successfully addressed your concerns or where it may have fallen short. In our second follow up we provided further clarifications and we are eager to hear your concrete reflection on them
> > > >
> > > > We look forward to your detailed feedback before the end of the discussion period.

---

> > ### Author Response · Authors · 2025-08-06
> > **Kindly checking-in again ....**
> >
> > Dear Reviewer,
> >
> > Once again, we thank you for your review and your subsequent acknowledgment of our rebuttal.
> >
> > We are writing to you to respectfully inquire whether our most recent response has sufficiently addressed the concerns you outlined. We are still eager **to receive more concrete feedback** on how our rebuttal and subsequent clarification have addressed your points and where they may have fallen short in terms of clarification.
> >
> > Thank you for your time and consideration, and we hope to hear from you before the end of the discussion period

---

> > > ### Comment · Reviewer_cdP4 · 2025-08-07
> > >
> > > Thanks for the author's rebuttal. I think I have expressed that my concerns have been addressed, and I will raise my score. Additionally, I think the dataset proposed by the authors to be meaningful and valuable for advancing research in this area.

---

### Official Review · Reviewer_ffWR · 2025-06-29

**Rating:** 5
**Confidence:** 3

**Summary:**

This paper focuses on the *pluralistic alignment* of T2I safety evaluation, where *pluralistic alignment* refers to that an image is evaluated in a way that reflects the diverse (even confliciting) perspectives and values of different demographic and cultural groups. To this end, this paper introduces DIVE, a dataset consisting of 1,000 prompt-image pairs sampled from Adversarial Nibbler and annotated by 637 human raters across 30 intersectional demographic trisections (gender × age × ethnicity).  The authors present empirical analyses demonstrating how demographic factors influence harm perception and how current safety classifiers and policy-based raters often miss biases that diverse human raters detect.

**Additional Feedback:**

How do the authors envision this dataset being used in practice to improve T2I model safety? Are there concrete plans or tools for leveraging the rich demographic annotations, for example, fine-tuning safety classifiers, or safety alignment in T2I models?

**Dataset Code Accessibility:**

Yes

**Ethical Considerations:**

No, there are no or only very minor ethics concerns

**Final Justification:**

Fully address my concerns.

**Limitations Weaknesses:**

1. While the DIVE dataset is methodologically impressive and reveals critical gaps in current safety evaluation, the paper stops short of demonstrating how this data can concretely be used to mitigate harmful generations in T2I models. Without a clear pipeline showing how pluralistic human feedback translates into improved model behavior, the link between data collection and alignment improvement remains aspirational rather than operational.
2. DIVE dataset consists only 1,000 prompt-image pairs, which is relatively small compared to the scale of modern generative model evaluation.

**Strengths Contributions:**

1. The pluralistic alignment is a growing concern in AI safety that is underexplored in current datasets and benchmarks.
2. The authors carefully construct a demographically diverse rater pool with uniform coverage across 30 intersections, which is rare and methodologically sound.
3. Empirical findings reveal substantial divergence in harm perception across demographic lines, challenging monolithic safety standards.

---

> ### Author Rebuttal · Authors · 2025-07-30
>
> Thank you for recognising the importance and the methodological endeavour of this work, we appreciate your recognition of the critical gaps in safety evaluation DIVE showcases. Thank you also for providing further feedback to help us frame our results and overall message as clearly as possible. Below we provide a three-part response to your questions and concerns:
>
> ## Reviewer comment 1: Importance of DIVE and analysis for pluralistic T2I alignment and concrete plans and tools for leveraging DIVE
>
> Thank you for this critical feedback. We make four points to address this concern. We **(I)** provide the main points showcasing importance of DIVE and the analysis for T2I alignment, **(II)** we contextualise the work in terms of the goals of the D&B track, **(III)** we propose edits to the framing of the paper to clarify the message, and lastly **(IV)** we provide new experiments to supplement the point in Section 5.2 regarding steerability of LLMs via DIVE.
>
> (I) The DIVE dataset and analysis provided in the paper is crucial for pluralistic alignment of T2I models as it showcases **three main points:**
> - Current safety evaluation approaches using automatic safety evaluators and policy-driven human raters and baseline prompted LLMs **do not capture the perspective on safety of the real-world diverse user base**. Such a diverse perspective is non-trivially different from the status quo.
> -  There are **important nuances in diverse people’s perception of safety of T2I model outcomes** based on their demographic identity. Identifying such nuances is novel and has not been shown before. A key question on the path towards pluralistic alignment, is how to operationalise pluralism in the context of safety: In this direction, some results we show are demographically intersectional groups are more cohesive and, and different intersectional groups identify harms differently in different prompt-Image types.
> - Lastly our dataset and experiments with Gemma, which we have expanded in the rebuttal to include a larger model Gemini 1.5 Flash (results of new experiments at the end of the response), show that LLM-based judges for safety evaluation with demographic data and few-shot examples show modest correlations with diverse raters’ scores. Thus, further demonstrating the gaps and available headroom in moving towards pluralistic alignment of T2I models.
>
>
>
> **(II) Alignment with D&B track goals:** We would like to note (lines 319-333) that our DIVE dataset contributes in the same space as prior research in text-to-text datasets, published in the NeurIPS Dataset & Benchmarks track. These works similarly identified gaps in safety evaluations and published a dataset that can be used to improve model alignment (without providing a model-based solution to alignment). Examples of such papers are
> - “The PRISM Alignment Dataset” by Kirk et al, 2024 (best D&B paper in 2024) and
> - “DICES Dataset: Diversity in Conversational AI Evaluation for Safety”, by Aroyo et al 2023.
>
> Both these papers focus on text-to-text issues, and our dataset builds upon them with more advanced statistical design and uniquely rich data into the text-to-image space. Furthermore, the D&B track goals are to stimulate research **papers introducing datasets that help evaluate and benchmark models**. Similarly, the scope of DIVE is to show the variety of gaps and issues caused by not considering diverse viewpoints crucial to be addressed as part of pluralistic alignment.
>
> **(III) Updates to paper based on feedback:**
>
> - **Framing**: We will clearly frame the narrative to clarify that the point you raised **is not an oversight but an important insight aligned with the goals of the paper**:
>     - The **core contribution of this paper is the curated, pluralistic dataset**, intended to provide a valuable benchmark for the research community.
>      - Our **experiments serve as crucial analysis** to demonstrate the dataset's utility and provide a necessary "headroom analysis" of current model capabilities
>
> - **Future work section**: Lastly, following your feedback, in the future work section we will outline clearly the **specific next steps towards operationalizing this dataset for improving pluralistic alignment in T2I models**:
>      - DIVE can be used for model training and fine-tuning and evaluating LLMs’ ability to replicate diverse safety perspectives, as has been done in past work building upon T2T safety datasets [3,4].
>      -  DIVE provides rich preference data via granular scoring which is useful for RLHF-based experiments on steering models towards alignment with diverse raters [5], which is a large and open research problem. Many approaches in reward modeling [6] and mixed experts [7] would benefit from the diverse viewpoints in DIVE.
>
> We will clarify the message and the narrative of the paper to emphasise and concretely specify the role of the dataset and the following experiments and analysis in the Introduction and Conclusion section.
>
> **(IV) New experiments for LLM-based raters with Gemini 1.5 Flash**
>
> Based on general reviewer feedback, we conducted **new experiments with a larger model family** (Gemini 1.5 Flash distilled from a 200B parameter Gemini 1.5 Pro model) for each strategy - (1) prompting with demographic information and (2) prompting with demographic information plus few shot examples from DIVE.
>
> _Our new experiments with a larger model show correlation improvement with few shot examples, showcasing steerability._
>
> We now provide the Kendall-Tau correlation of the diverse raters and the LLM-based rater averaged for each top level demographic (in parenthesis):
>
> **(1) Gemini1.5F prompted with demographic information (extension of Table 2 in paper)** KT correlation:
> - 0.25 (Men), 0.26 (Women),
> - 0.25 (GenX), 0.25 (Millennial), 0.26 (GenZ),
> - 0.23 (Black), 0.24 (Eastasian) 0.26 (Latinx), 0.27 (Southasian), 0.27 (White)
>
> **(2) Gemini1.5F prompted with demographic information and few-shot examples**
> KT correlation:
> - 0.28 (Men), 0.29 (Women),
> - 0.28 (GenX), 0.28 (Millennial), 0.30 (GenZ),
> - 0.27 (Black), 0.27 (Eastasian) 0.28 (Latinx), 0.30 (Southasian), 0.30 (White)
>
> **Insights:** We see that the larger model when provided demographic information and few-shot examples performs better than the other model (Gemma) and other strategies considered. Interestingly, **the correlation improves with few shot examples for Gemini, showing steerability of larger models**. However, the correlations remain modest, thus emphasising the importance of DIVE for future research on methods to develop LLM-based raters in the T2I space. We will discuss this result in further detail in the paper breaking down the results by trisections, to convey the extent of the gap in representation of different identities with LLM-based approaches to evaluation.
>
>
> ## Reviewer comment 2: DIVE dataset is relatively small compared to the scale of modern generative model evaluation.
>
> **Depth vs breadth trade-off:**
>
> - The DIVE dataset has a unique focus on understanding the safety perspectives of 30 unique demographic trisections on each PI pair. Thus, we take a depth-first approach to safety evaluation by trading off a higher PI-pair count in favor of ratings from more unique demographic identities.
> - The importance of this is evidenced by the findings in Section 4 and 5, where we see that some demographically intersectional groups are more cohesive, different intersectional groups identify harms differently in different prompt-Image types, and current safety evaluation methods show variance in their representativeness of different demographics.
> - The high number of ratings enable statistically rigorous analysis, as previously discussed in [1].
> - In addition, past research dataset [2] for pluralistic alignment with a focus on diverse annotation has a similar dataset size.
>
>
> **Diversity within Prompt-Image-pair set:**  As described in Section 3, when choosing the 1000 PI-pair set, several important considerations were taken into account to ensure that the PI pairs are diverse and showcase a variety of harms and are suitable for understanding pluralistic alignment. **This ensures that the dataset has useful and actionable implications for safety evaluation**. We list the considerations here:
> - The dataset is diverse containing three different harm types each with sizeable representation
> - The dataset contains PI pairs concerning unique and varied topics each with uniform representation
>
> Lastly, model evaluation datasets are typically optimized (ranging about 1-2K items) for optimum coverage across issues and policies, and typically are kept small as evaluations need to be performed often and larger datasets increase effort and turn around time.
>
> **References:**
>
>
> [1] How Many Raters Do You Need? Power Analysis for Foundation Models. Homan et al 2023
>
>
> [2] DICES Dataset: Diversity in Conversational AI Evaluation for Safety. Aroyo et al 2023.
>
>
> [3] How Far Can We Extract Diverse Perspectives from Large Language Models? Hayati et al 2024
>
>
> [4] LLMs instead of Human Judges? A Large Scale Empirical Study across 20 NLP Evaluation Tasks. Bavaresco et al 2025
>
>
> [5] Personalizing Reinforcement Learning from Human Feedback with Variational Preference Learning. Poddar et al 2024
>
> [6] MaxMin-RLHF: Alignment with Diverse Human Preferences. Chakraborty et al 2024
>
> [7] Collab: Controlled decoding using mixture of agents for llm alignment Chkraborty et al 2025

---

### Official Review · Reviewer_W1jd · 2025-07-02

**Rating:** 5
**Confidence:** 3

**Summary:**

This paper introduces the DIVE dataset, designed for pluralistic alignment in text-to-image (T2I) safety evaluation. The dataset consists of 1000 prompt-image pairs spanning three violation types and twelve topics, each annotated for perceived harm by 637 human raters distributed across 30 intersectional demographic groups (trisections of gender, age, and ethnicity). The paper provides quantitative and qualitative analyses showing significant, context-sensitive divergences in harm perception tied to demographics, and compares pluralistically collected annotations to those from policy raters and automated safety tools. Additionally, it explores how such diverse annotations might be used to steer LLM-based evaluators toward more pluralistic judgements. The contribution advances data-centric approaches for fairness and social context in T2I model safety, situating the dataset and findings within related work and common evaluation practices.

**Dataset Code Accessibility:**

Yes

**Ethical Considerations:**

No, there are no or only very minor ethics concerns

**Final Justification:**

Considering that the dataset’s fine-grained categorization of diverse value groups contributes meaningful value to the field, I am inclined to recommend acceptance. The authors resolved most of my concerns during the discussion phase, so I have slightly raised my score while lowering my confidence level. Nevertheless, the paper still needs more discussion on how the data can be used in practice and, more broadly, on downstream application strategies.

**Limitations Weaknesses:**

1. Since DIVE's Prompt-image (PI) pairs are mainly derived from pre-existing open-source data, I am more curious about how the authors control the quality of the PI pairs used?

2. While DIVE is designed for alignment of T2I models, its experiments and analysis mainly focus on statistical results among different rater groups, and there are fewer analysis on how it affects the T2I model and how it helps the alignment of T2I models.

3. Section 5.2 and Table 2 examine LLM-raters using a single 4B-parameter open-source model (Gemma), and only report modest correlations even after prompt-based demographic steering. Assessing alternative model families, or fine-tuning with DIVE, could strengthen this point.

4. Although ethical considerations are broadly addressed, concrete discussion of potential negative impacts from dataset release (such as reinforcing demographic partitions or being gamed for unsafe model development) could be expanded.

**Strengths Contributions:**

1. Beyond binary or singular harm judgments, the dataset includes Likert-scale harm ratings (to self and others), harm type categorization, and optional qualitative reasoning (Figure 1b and Section 3.2), supporting richer and more granular analysis.

2. The empirical analyses, supported by Table 1 and Figure 3, go beyond surface demographic grouping and convincingly show that intersectional identities (e.g., GenZ-Black women) provide more cohesive and distinct harm perspectives than broad categories, an insight valuable for alignment research.

3.  Has a better demo of how to use DIVE data in the future. The systematic comparison with policy raters, LlavaGuard, and ShieldGemma (Figure 4) quantifies the concrete blind spots of current safety classifiers, especially on bias-type harms, and highlights the empirical value of pluralistic human feedback. Section 5.2 and Table 2 experimentally probe the ability of LLMs to mimic demographic group harm ratings, offering both a valuable resource and an empirical caution for future scalable evaluation pipelines.

4. The paper is very well-written and organized

---

> ### Author Rebuttal · Authors · 2025-07-30
>
> We are pleased that the reviewer found our paper insightful with respect to the intersectional perspective on safety and blindspots categorization and recognized the novelty of the dataset in terms of its richness and granularity and moving the status quo beyond binary judgments. Below we provide detailed responses to your questions:
>
> # 1: Quality control of the PI pairs used
>
> Thank you for raising this. Our selection process for the PI pairs was rigorous (details in section 3.1 on Dataset curation). We agree that prompt set quality is of high importance, and to address your question, we emphasize here the two main steps taken to ensure the quality of PI pairs used in DIVE and account for it, and the following points will be emphasized and clarified in the revised manuscript:
>
> - **Every PI pair underwent manual inspection** by at least one author to verify its suitability for inclusion in the present study. In this we also filtered out images flagged for low quality (such as images with unnatural human body representation unrelated to the prompt query)
> - To ensure that the quality of images does not interfere with the safety ratings of the study participants, **we instructed participants on how to incorporate image quality in their safety perception** and accounted for it in the response form. The full rater instructions will be added in the Appendix (A.2). We provide abridged instructions here:
>
> “**Note on image quality**:We ask you to NOT judge the safety of an image based SOLELY on image quality / imperfections. To evaluate an image, you should consider whether the image is UNSAFE even if the image imperfections were fixed, and mark it accordingly. If you are unsure about the safety of the image due to image imperfections, answer Q1 with "Unsure" and select “I could not make sense of the image, e.g. image quality was bad”.”
>
> Lastly, the **PI pairs were sourced from Adversarial Nibbler dataset** [1] which had three unique and important features making it **a high quality dataset**, well aligned with our research goals:
> - AdvNib Challenge participants were instructed to provide implicitly adversarial prompts, which resulted in a prompt set well positioned to uncover novel safety blindspots [1].
> - The corresponding images were generated using a diverse set of 6-9 text-to-image models ensuring coverage of model families.
> - Each PI pair in AdvNib was annotated by five policy raters as a verification of the pairs and annotations provided by challenge participants.
>
>
>
> # 2: How DIVE experiments and analysis help T2I model alignment
>
> The analysis provided in the paper is crucial for pluralistic alignment of T2I models as it showcases **three main points**:
>
> - Current safety evaluation approaches using automatic safety evaluators and policy-driven human raters and baseline prompted LLMs **do not capture the perspective on safety of the real-world diverse user base**. Such a diverse perspective is non-trivially different from the status quo.
> - There are **important nuances in diverse people’s perception of safety of T2I model outcomes** based on their demographic identity. Identifying such nuances is novel and has not been shown before. A key question on the path towards pluralistic alignment, is how to operationalise pluralism in the context of safety: In this direction, some results we show are demographically intersectional groups are more cohesive, and different intersectional groups identify harms differently in different prompt-Image types.
> - Lastly our dataset and experiments with Gemma, which we have expanded in the rebuttal to include a larger model (Gemini 1.5 Flash), show that LLM-based judges for safety evaluation with demographic data and few-shot examples show modest correlations with diverse raters’ scores. Thus, further demonstrating the gaps and available headroom in moving towards pluralistic alignment of T2I models.
>
> **Alignment with D&B track goals:** We would like to note (lines 319-333) that our DIVE dataset contributes in the same space as prior research in text-to-text datasets, published in the NeurIPS D&B track. These works similarly identified gaps in safety evaluations and published a dataset that can be used to improve model alignment (without providing a model-based solution to alignment). Examples of such papers are
> - “The PRISM Alignment Dataset” by Kirk et al, 2024 (best D&B paper in 2024) and
> - “DICES Dataset: Diversity in Conversational AI Evaluation for Safety”, by Aroyo et al 2023.
>
> Both these papers focus on T2T issues, and our dataset builds upon them with more advanced statistical design and uniquely rich data into the T2I space. Furthermore, the D&B track goals are to stimulate research **papers introducing datasets that help evaluate and benchmark models**. Similarly, the scope of DIVE is to show the variety of gaps and issues caused by not considering diverse viewpoints crucial to be addressed as part of pluralistic alignment.
>
> **Updates to writing**
>
> - **Framing**: We will clearly frame the narrative to clarify that the point you raised **is not an oversight but an important insight aligned with the goals of the paper**:
>    - The **core contribution of this paper is the curated, pluralistic dataset**, intended to provide a valuable benchmark for the research community.
>     - Our **experiments serve as a baseline analysis** to demonstrate the dataset's utility and provide a necessary "headroom analysis" of current model capabilities
>
> - **Future work section**: Lastly, following your feedback, in the future work section we will outline clearly the **specific next steps towards operationalizing this dataset for improving pluralistic alignment in T2I models**:
>     - DIVE can be used for training and fine-tuning and evaluating LLMs’ ability to replicate diverse safety perspectives, as has been done in past work building upon T2T safety datasets [2,3].
>     - DIVE provides rich preference data via granular scoring for RLHF-based model steering towards alignment with diverse raters [4], which is a large and open research problem. Many approaches in reward modeling [5] and mixed experts [6] would benefit from the diverse viewpoints in DIVE.
>
>
> # 3: Concerns with use of 4b model for LLM-based autoraters
> Thank you for raising this. To have a more comprehensive evaluation, we conducted **new experiments with a larger model family** (Gemini 1.5 Flash distilled from a 200B parameter Gemini 1.5 Pro model) for each strategy - (1) prompting with demographic information  and (2) prompting with demographic information plus few shot examples from DIVE. Here, we provide the Kendall-Tau correlation of the diverse raters and the LLM-based rater averaged for each top level demographic (in parenthesis):
>
> **(1) Gemini1.5F prompted with demographic information (extension of Table 2 in paper)** KT correlation:
> - 0.25 (Men), 0.26 (Women),
> - 0.25 (GenX), 0.25 (Millennial), 0.26 (GenZ),
> - 0.23 (Black), 0.24 (Eastasian) 0.26 (Latinx), 0.27 (Southasian), 0.27 (White)
>
> **(2) Gemini1.5F prompted with demographic information and few-shot examples** KT correlation:
> - 0.28 (Men), 0.29 (Women),
> - 0.28 (GenX), 0.28 (Millennial), 0.30 (GenZ),
> - 0.27 (Black), 0.27 (Eastasian) 0.28 (Latinx), 0.30 (Southasian), 0.30 (White)
>
> **Insights:** The larger model when provided demographic information and examples performs better than the smaller model. Importantly **providing examples from DIVE improves the metric with the Gemini model, showcasing steerability.**
>  However, the correlations remain modest, emphasising the importance of DIVE for future research on methods to develop LLM-based raters for T2I. We will discuss this result in further detail in the paper breaking down the results by trisections, to convey the extent of the gap in representation of different identities with LLM-based raters.
>
> **Scope of the paper:** we would like to reiterate here that **LLM-as-judge style model alignment via algorithm design and experimentation is out of scope for this work**, in keeping with the D&B track goals, and outlined as future work in our manuscript. We will revise Section 5.2 to clarify this goal and focus the narrative on steerability of LLMs using off-the-shelf methods. In the future work section, we will encourage further experimentation for steering LLM-raters with DIVE.
>
> # 4: Additional ethical considerations
> To address concerns of dataset release, we note a few related points
> - **Low risk of dual-use:** numerous datasets of this type already exist (over 150 adversarial datasets and over 100 different stereotypes and demographics datasets) highlighting LLM fairness and safety issues.
> - **Disturbing images safeguard:** Our approach includes a critical, practical safeguard: the DIVE dataset release does not include the images, only the prompts and annotations. The images themselves will be available only upon request, which will allow our team to maintain oversight on their use.
> - **Reinforcing demographic partitions:** our data collection methodology was deliberately chosen to favor an intersectional view of identity, building on practice in related socio-technical research, which directly counters the creation of negative stereotyped groupings
>
> To address your feedback, we will **extend the ethical considerations section** to clarify the points above.
>
> **References**
> [1] Adversarial Nibbler: An Open Red-Teaming Method for Identifying Diverse Harms in Text-to-Image Generation. Quaye et al 2024
>
> [2] How Far Can We Extract Diverse Perspectives from Large Language Models? Hayati et al 2024
>
> [3] LLMs instead of Human Judges? A Large Scale Empirical Study across 20 NLP Evaluation Tasks. Bavaresco et al 2025
>
> [4] Personalizing Reinforcement Learning from Human Feedback with Variational Preference Learning. Poddar et al 2024
>
> [5] MaxMin-RLHF: Alignment with Diverse Human Preferences. Chakraborty et al 2024
>
> [6] Collab: Controlled decoding using mixture of agents for llm alignment Chkraborty et al 2025

---

> > ### Comment · Reviewer_W1jd · 2025-08-03
> >
> > Thank you for your response. I understand that alignment methodology is not the primary focus of this paper. However, from another perspective, demonstrating how this dataset can be used to improve the alignment performance of T2I models is an important piece of evidence for the dataset’s value. Therefore, I recommend adding more relevant discussion on this point.
> >
> > The authors have addressed most of my concerns, and I will raise my score accordingly.

---

### Note · Authors · 2025-08-13

**Summary of paper contributions and strengths as articulated by reviewers**
First, the paper introduces DIVE, a dataset praised as "groundbreaking" (323P),"rare and methodologically sound" (ffWR), and “more sophisticated than typical high-level groupings” (cdP4) for evaluating T2I safety from diverse, intersectional perspectives. Secondly, the dataset contains rich, granular feedback enabling nuanced analysis (W1jd). Next, it provides "robust" and "compelling" empirical evidence that harm perception is not monolithic and varies significantly across intersectional identities (323P, cdP4, ffWR, W1jd). Finally, the work demonstrates its practical value by identifying "concrete blind spots" of current safety tools (W1jd, cdP4), and offering a baseline for steering LLMs toward more pluralistic judgments using DIVE. (W1jd)

**Improvements based on feedback**
- Addressing reviewers, we conducted new experiments with larger models to show the steerability of LLMs-as-judges using the DIVE dataset, thus providing more evidence for its usefulness for pluralistic alignment in T2I. Based on this addition, reviewers 323P, cdP4 and W1jd indicated their concerns were addressed and raised their scores. Reviewer ffWR noted a similar limitation but was unresponsive during the discussion period.
- We also clarified the narrative based on reviewer feedback to emphasise the importance of the dataset and the analyses towards pluralistic alignment.

**Unresponsive ethics reviewer**

We responded to the ethics concerns raised by reviewer RKjp however we didn’t get a confirmation despite our efforts to engage with them. It is important to point out that RKjp's concerns are contradicted by the positive remarks by other ethics reviewers (3F7V, 4b4N).

- **Reviewers 3F7V, 4b4N confirmed we took appropriate and comprehensive measures to mitigate risks**, such as a protocol approved by an Internal Review Board (IRB) (e.g. with opt-in checks, content warnings, image blurring button, mental health resources, and the option to skip items or withdraw, encouraging taking breaks (and accounting for it in compensation).
- We also provided clarification on each of the reviewer’s concerns including details about IRB, rater compensation and well being-practices

Finally, we thank all reviewers and the AC for their continuous engagement and insightful comments which have improved the work immensely. We are grateful for all the reviewers’ and ethics reviewers’ positive assessment of our work!

---

### Decision · Program_Chairs · 2025-09-18

**Decision:**

Accept (spotlight)

**Comment:**

This paper introduces DIVE, a dataset for evaluating pluralistic safety in T2I models which spans 1,000 prompt-image pairs annotated by 637 raters across 30 intersectional demographic groups. All four reviewers reached a positive consensus on the paper (4x accepts with 1x 2/5 and 3x 3/5 confidence), citing the dataset’s richness for studying pluralistic views on image-prompt safety. The paper's main finding - that what is considered a 'safe' model output by one person may not be considered safe by others - is novel and important for the broader research community to acknowledge and embrace in evaluations goings forwards.

The main concern raised by multiple reviewers was the lack of empirical evidence showing how the dataset improves model alignment, despite this being a central framing of the paper (including in the title). I believe, however, that (i) the work, in its current shape, aligns with the D&B track’s scope, where similar dataset-focused contributions have been accepted, and (ii) there is high value in datasets with high-quality granular human annotations, especially in nuanced areas like model safety. Three ethics reviews were also conducted, and all issues were acknowledged and addressed by the authors.

For the camera-ready, I recommend the authors (i) reframe the narrative away from alignment and more toward the dataset’s contributions and analyses, and (ii) include concrete future directions for using DIVE to support pluralistic model alignment. Despite the noted limitations, this work will serve as a valuable resource for safety evaluation in text-to-image models going forward.